# Coping and Resilience Strategies among Ukraine War Refugees

**DOI:** 10.3390/ijerph192013094

**Published:** 2022-10-12

**Authors:** Lluis Oviedo, Berenika Seryczyńska, Josefa Torralba, Piotr Roszak, Javier Del Angel, Olena Vyshynska, Iryna Muzychuk, Slava Churpita

**Affiliations:** 1Faculty of Theology, Pontifical University Antonianum, 00185 Rome, Italy; 2Faculty of Theology, Nicolaus Copernicus University, 87-100 Torun, Poland; 3Theological Institute Murcia, 30003 Murcia, Spain; 4Independent Scholar, Ukraine

**Keywords:** refugees, trauma, resilience, coping, altruism

## Abstract

(1) Background: The war in Ukraine has triggered a huge humanitarian crisis: millions of refugees have escaped from their homes looking for shelter beyond Ukraine’s borders. This emergency offers a unique opportunity to investigate and to document those characteristics of the human condition that emerge in exceptional circumstances provoked by war. Based on considerations derived from recent studies on resilience, the research at the base of this paper was conducted to better understand the circumstances, states of mind, and coping mechanisms of the refugees leaving their homes looking for security. (2) Methods: A qualitative approach was applied, resorting to semi-structured interviews (n = 94) to closely explore the traumatic experiences lived by refugees and to identify their coping strategies. (3) Results: The data obtained pointed to a plurality of coping and resilience strategies. Maintaining communication with separated loved ones as well as experiencing accompaniment by helpers and hosts emerged as principal elements for coping and resilience. It was found that a prior development of interior life or practice of prayer served as psychological “capital” that increased their resilience. (4) Conclusion: The results point to the need to care for social networking, attention by hosts, and cultivating interior life as keys for resilience.

## 1. Introduction

### 1.1. Studying Resilience among War Refugees

The terrible war taking place in Ukraine since the end of February 2022 has provoked a huge exodus of civilians looking for security amid the threats of bombings, destruction, and invasion-related hazards to life. The substantial population displacement from endangered areas has engendered a humanitarian crisis of vast proportions. Many countries in Europe have strongly engaged in efforts to offer shelter and help for the urgent needs of those who have fled their homes. Indeed, the war has revealed the worst and the best of humans—whether they are directly or indirectly involved.

Since the first weeks of the conflict and after witnessing the scale and reach of this crisis, an international team of researchers (which has previously collaborated in projects on coping, religion, and related dimensions) decided to study the elements of adversity, stress, coping, and resilience manifest in this refugee crisis. The principal objective was to better understand coping and resilience among those most severely affected by this war. This project was not motivated by mere curiosity, but by an interest in helping others who find themselves now or in the future in similar situations and who have to cope with such struggles. A normative and even therapeutic intention informed this entire project.

Besides the general objective, we conceived other objectives for our projected research:To assess coping strategies and resilience among war refugees (considering age, education, socio-economic background, etc.)To identify the influence of religion on the coping mechanisms, along with other strategies aimed to achieve resilience.To ascertain motivations among volunteers (both formal and informal) and those engaging in hosting refugees.To increase our comprehension of the dynamics of the human condition in times of great struggle (such as war), and how people manage to survive and to adapt in those harsh circumstances.To identify and describe altruistic attitudes and their special features arising in time of great stress and demand for help.To identify and document experiences and practices that could be included in the educational curriculum of young generations regarding both the positive and negative aspects of human nature.To contribute to and complement the development of both past and current studies on humans flourishing in hard conditions.

The present project builds upon an ample repertoire of published studies, with similar aims and results. Our own study aims to contribute with new data to the available studies and to explore novel or different means these vulnerable populations achieve resilience.

### 1.2. State of Research on Refugees and Resilience

When dealing with our subject, at least three different but often related research areas can be identified: (1) studies on coping and resilience; (2) studies on refugees, their mental health and wellbeing; and (3) studies on religion and its role in coping and resilience. These areas contain a consistent body of published research, and offer points of convergence relevant to refugees, and their coping and resilience, with attention to religious factors.

The exhaustive bibliographic database PubMed can provide an initial orientation. In it, there are 155.508 entries on “coping strategies”: more specific entries on “religious coping” amount to 3.041. Searching for the word “resilience” yields 55.892 entries; “resilience and religion”, 728; “resilience and refugees”, 486; and “refugees and mental health”, 3573 (data from 2 August 2022). More specifically, on “refuges and resilience”, the PubMed database offers 503 entries; adding the filter “religion”, we obtained 22 entries. To identify the most relevant published papers that would offer useful information to better contextualize our own research, we traced a path from the literature with the most general focus to that with the most detailed focus: centred on refugees, their struggles, and their coping and resilience strategies.

It is convenient to start with some broad views and definitions of resilience. The Merriam-Webster Dictionary offers the following: “an ability to recover or adjust easily to misfortune or change”. This ability is closely related to the capacity to cope, or—again with the same dictionary—“to deal with and attempt to overcome problems and difficulties”. Indeed, one finds definitions that include coping as part of a broad view on resilience: “A general and generic property of systems, the broad ability of a system to cope with disturbances without changing state” [1]. The authors of this latter study specify 22 uses or meanings of “resilience”, even if their focus falls more on the ecological sciences. Possibly, the following definition is more operative in our case and context: “A capacity to confront, absorb, withstand, accommodate, reconcile, and/or adjust to condition of adversity, setback, and challenge in the pursuit of desired or desirable goals or states” [2].

Going a little deeper into the topic of resilience, one finds that most studies on resilience build upon a wide network of existing uses and applications, in a very transdisciplinary zone. Indeed, resilience has now become a hub for many crossing and interacting sciences and approaches, including biology, ecology, social sciences, and therapeutic sciences. In several cases, the emphasis is on the systemic character of resilience. After undertaking a multidimensional review, one paper was found to identify seven common traits; resilient systems are characterized by: (1) the ability to deal with adversity; (2) a processual character; (3) the ability to make expected trade-offs; (4) open, dynamic, and complex architectures; (5) promotion of connectivity; (6) the capacity to learn; and (7) diversity, redundancy, and participation. All this is aimed at “securing the resources required for sustainability in stressed environments” (Ungar 2018, 1). An alternative view, aimed at a therapeutic context, builds on a selective review of published studies in several disciplines, and points to three big themes: “(a) hardiness strengthens the ability to harness resources, (b) regulatory flexibility fosters positive functioning, and (c) challenges enhance the ability to rebound” [3].

The former approaches may still appear as too generic and abstract when dealing with situations of concrete personal and social stress, but they provide a useful framework when trying to cope with such crises and struggles. By summarizing the many approaches and descriptions, one sees that: (1) resilience is a common property of many systems: physical, living, personal and social; (2) resilience makes these systems especially resistant before adversity, big changes or crises; and (3) resilience works through a variety of dynamics that make such systems more adaptive. Obviously, though, it is hard to transfer traits that ensure resilience in physical systems (or even in biological or economic systems) to the systems required for a person to overcome a major difficulty or stress. Recent research based on field work has tried to identify internal and external sources of resilience [4].

This is why it is important to take another step deeper and to identify the stressors and challenges that afflict, in a specific way, refugees. Many studies provide good descriptions on these issues. A systematic review from 2005 described the mental disorders that plague many refugees, identifying post-traumatic stress disorder (PTSD) and major depression as the main problems suffered by those fleeing their countries in the context of war [5]. More recent studies identified similar problems: a systematic review on studies on children and young refugees found a significant impact of PTSD, depression, anxiety, and emotional-related problems [6]. An empirical study on refugees in Albania reports an incidence of PTSD and “psychiatric morbidity” in more than 30 percent of the surveyed sample [7]. Some forms of psychological distress were identified in another recent empirical study on refugees in Germany [8].

A final step is to look for strategies or ways to cope and gain resilience in this severely stressed population. Many studies have tried to better understand how refugees manage to overcome their problems and to live good lives after they have resettled. A systematic review of published literature on resilience strategies by young refugees identified six relevant factors contributing to their resilience: (1) social support, (2) acculturation strategies, (3) education, (4) religion, (5) avoidance, and (6) hope [9]. A study from the same authors gathering data from a rather small group (n = 16) of young refugees in the Netherlands reveals four strategies: acting autonomously, performing at school, perceiving support from others, and participating in the new society [10]. More recent empirical research has shown a moderately positive correlation between resilience and intrinsic religiosity among young refugees in Malaysia [11]. This result is confirmed by another recent and extensive systematic review and meta-analysis, summarizing 34 observational studies and revealing a moderate correlation between religion/spirituality and resilience defined as “the ability to recover or cope with adverse situations” [12]. Other studies highlight the positive role religion plays in that process [13,14,15,16,17,18]. Usually, these studies relate to displaced populations from poor areas and diverse religious backgrounds. Almost no research has focused on Western and more affluent populations with Christian majorities.

Other studies highlight the role of agency as a key to resilience: refugees endowed with greater sense of agency experience higher resilience levels [19]. However, other recent research points to contextual factors contributing to refugees’ wellbeing and adjustment to such external factors [20]. A recent paper calls for professional interventions to assist Ukraine refugees in their several psychological difficulties [21]. Furthermore, an expert in resilience studies—who even speaks about a “science of resilience”—after reviewing more articles on this topic, claims that the capacity for resilience results from a complex and systemic interaction between environmental and internal or psychological resources, and less from any unique factor [22]. Both the examined literature and the evidence clearly point to the central role of external and internal social support for those who suffer war and displacement [8,13,15,19].

This short review of the literature provided some guidelines to better frame our own research. It is apparent that refugees suffer different forms of distress, as expected from people traumatized by the experience of war, displacement, and resettlement in unfamiliar environments. It is clear, too, that most of them develop forms of resilience or activate processes to adapt to their new situation and to cope with the adverse conditions they have left behind and that are still afflicting their many relatives and friends. The available literature has identified several forms or dimensions that can be recognized as resilience strategies, including religious faith and practices. However, several voices have complained about the scarcity of studies that (1) better explore those pending issues and (2) engage more in field work to observe how refugees deal with their problems and gain resilience [14,23].

Our team has tried to add more data and analysis to the currently available information to depict a more accurate panorama amid the current international emergency involving Ukraine war refugees. To that end, our team has conducted a significant number (n = 94) of semi-structured interviews in four different locations in three countries to gather relevant information, especially regarding the following five crucial issues: (1) what are the main stressors that are afflicting Ukrainian refugees; (2) what are the main coping strategies that they practice and advise for others; (3) what is the image they have of the hosts and other people they have met; (4) what are their current state and expectations; and finally, (5) what role does religion and religious prayer play in their stressing context. We will describe first the methods adopted and move later to an extensive account of the main results from this qualitative survey.

## 2. Materials and Methods

Our team designed this research project a few weeks after the start of the Ukraine war (24 February 2022) and after becoming aware of the great flux of refugees across different European countries.

The formerly described objectives are certainly ambitious and we tried to give priority in our research to identifying the positive values in what is a difficult situation for many people. For this reason, the present paper will deal just with the attitudes and experiences of refugees, and their struggles and resilience. Later, we would like to study the attitudes and experiences of those engaging as helpers and offering accommodation to fleeing persons.

This study, with its ethical protocols, was submitted to the Institutional Review Board of Antonianum University. Prior to this, several ethical issues were identified by our team members concerning the sensitive situation of refugees and how disturbing or unsettling the designed questions and interview could become for persons who are still traumatized by the recent war and their memories of having to flee. Our rationale was that such a study should be carried out to better assist other people in similar conditions. The project was approved, and all the ethical issues have been addressed. Indeed, the team decided that the interviews should be conducted only by other Ukraine refugees in their same language and after informed consent. Emotional issues related to the experience of war, which are also shared by the interviewers, have been considered. The interviewers were selected among Ukraine war refugees, who because of their ethnic affinity with the interviewees and sharing similar experiences, could have an advantage to empathise with them, and thus to be in a better position to address the stress and emotional trauma that could emerge during the interviews. This is particularly noticeable in the selection of two Ukrainian psychotherapists in Poland, a country closest to the war experience.

After considering different options, we decided that the most fitting analytical approach to meet the designed objectives was a qualitative one, with a semi-structured interview as the best means to obtain relevant information. The questionnaire that served as the basis for the interview comprised four demographic questions (age, gender, education and religious affiliation) and nine open questions (including about travel experience and the people in the company of those fleeing; about people who helped; about coping strategies; about current feelings and expectations; and about religion and prayer). An additional questionnaire was designed for helpers or volunteers. Our target was to gather at least 100 interviews.

We looked for different locations and we recruited several Ukraine war refugees with enough experience, linguistic skills, and psychological training to conduct the interviews. In Poland, two women in two different localities (Wroclaw and Łodz) collected 38 and 20 interviews, respectively. In Rome, another Ukrainian woman collected 25 interviews. In Murcia (Spain), another Ukrainian woman collected 12 interviews; and a local team collected an additional 10 interviews from volunteers. The interviewers were selected considering their studies and experience in the psychological field. In all, the team gathered 94 completed interviews from refugees and 10 from helpers. Interviewers were compensated financially by a fixed amount for each completed interview. The same persons conducting the interviews were requested to record them and to translate them into English. Refugees were interviewed in a period between one and three months after they left Ukraine. The subjects were found in usual places where Ukrainian persons gather or go to seek information or assistance. A significant number of interviews was collected in a hotel near Wroclaw hosting about 300 refugees; a lesser number was collected by telephone from refugees living in other European countries. Interviews lasted an average of 1–3 h each.

A member of our team later arranged all this material into a dataset using the program MAXQDA 2022. This is a qualitative data analysis and mixed methods software. Codes were first created and programmed, then linked to the corresponding text segments. The result was a code-ordered table containing the extracted text segments. Using the program facilitated and significantly shortened the whole process of working with the texts and made a smooth transition to the subsequent SPSS statistics analysis activities possible.

The next step was to codify the answers into a SPSS dataset to reduce and better calculate the answers that were suitable for such a reduction and coding. After obtaining all the text segments we could easily compare the different answers and their respective tones and emphases. The analytical technique resorted to textual hermeneutics based on the textual evidence, main references and the meaning we could extract from each testimony, which helped to re-code those expressions into simplified scales. We admit to some degree of flexibility in doing that operation, but in most cases the coding was direct, immediate and clear. We applied a hermeneutic approach to content analysis, based on the immediate and evident meaning of the reported experiences.

For instance, the levels of distress suffered during travel while fleeing were codified using a scale of three levels: very stressful, quite stressed, and low stress. Through a textual hermeneutic procedure, our approach identified these levels of analysing words and expressions. We did not rely on text mechanic analysis but rather on the often-clear descriptions and meaning of the collected testimonies. The same scale was used for differentiating the current state or feelings of the subjects. The many coping strategies were simply recorded and organized to discover convergences, and a total of 20 possible answers were identified. Then the collected testimonies were carefully read to recognise patterns and to answer the main research questions of this study: on the initial experience and struggles of refugees; their coping strategies; their relationship with host and helpers; their current situation and expectations; and the role played by religious factors. Interviews were read, transcribed, coded, and each code was counted by reading the transcripts. Then, excerpts (testimonials) from the transcripts were selected to indicate the presence of the code to support the section of the code.

## 3. Results: The Process of Resilience from Initial Stress to Coping

As has been already described in the literature review, resilience is basically a process that requires some type of coping and adaptation. It is important to relate to the initial state, which can be identified by the memories the refugees kept about the situation that motivated their decision to leave their homes, and the uncertainties and struggles to find shelter and a way to live in a new and unfamiliar environment.

A first look at the demographic data of our sample is revealing: 79.8% are women, as can be expected since most men were not allowed to leave the country and were conscripted. The mean age is 46 years, with a standard deviation of 14.8. The education levels are high: completed high school (10.6%); university graduates or similar (51%); master level or double degree (20.2%). The religious affiliations are: Christian Orthodox (52.1%); Catholic (9.6%); other religion (10.6%); Protestant (1%); no religion (10.6%). As can be expected, most refugees profess the religious faith proper of their context: Orthodoxy; the rest can be considered as religious minorities. In any case, the low rate of religiously unaffiliated people in this sample was surprising (just 10.6%).

### 3.1. The Travel Experience

After reading the interviewed experiences, we coded them into three major groups: “easy”, “quite stressful”, and “very hard”. The table of frequencies tells us that that experience was very hard for almost 60% of refugees, quite stressful for 19%, and easier for the other 19%. Table 1 shows the composition of the families leaving their homes:

However, one must consider the oral stories gathered to obtain a glimpse of the great suffering that the war and invasion provoked. In many cases, people decided to flee after being under shelling for days or even weeks. They tell about their travel hardships: being sometimes close to bombing zones or rail stations under attack for several days; not being able to sleep; being under very cold temperatures; and experiencing scarcity of food. Many remember the numerous checkpoints they needed to cross in Ukraine, and the long queues and hours of waiting to pass through the border. Moreover, among all this, the great uncertainty about their immediate future. One must consider that only a proportion of those people had contacts in the countries they were trying to reach; just 11.7% were being assisted by friends or relatives, although over 77% expected to find a better place to stay than the places they left behind, places often characterized by bombing and other great threats. In many cases, the greatest concern of refugees was for their children, the effort to prevent them from experiencing the trauma of war, or simply ensuring their survival and security. In several cases, the answers reflect significantly negative psychological states—trauma, depression, and anxiety—in the same way that the literature cited previously has described.

Two testimonies from young mothers are very representative of those difficulties experienced by most refugees:

“Overall, it was very difficult. We were traveling by car and were stopping at different petrol stations. Almost every station had no electricity because they were afraid that they can be bombed. My husband was driving me and our two kids to the border. We have crossed the pedestrian border. On the day we arrived at the border the queue was 30,000 of those who wanted to cross. It was very cold, so my husband helped us to stay in the queue and cross the border. Even though we did not have much clothes, it was difficult for me as a woman to carry three bags and a small child. My son who is 6 years old also helped. The travel experience was difficult, dangerous and the overall atmosphere was frustrating as it was the beginning of March”.(29 years old, mother of 2 children, refugee in Poland)

“There was a constant feeling of fear, boundless concern for the life and health of my child. It is difficult for me to express everything. But the trip was a complete shock, it was very difficult to understand what to do, where to go. It was also important to me that the child did not hear my confusion. My husband could not go with us, I travelled without him for the first time. It also depressed me a lot. I wanted to finally get somewhere. At one point I just wanted to eat some hot soup. Strange feelings. I wouldn’t want to go through that again. When we arrived in this place, we were given a room with a toilet and a shower, fed, I even cried, I could not contain my emotions”.(35 years old, mother of one daughter, refugee in Poland)

### 3.2. The Coping Strategies

Our questionnaire contained two open questions related to coping: “What helps you to cope with this difficult situation?” and “What else helps to cope?” The second question was intended not just as repetition, but to solicit advice the interviewee might offer after his or her own experience to help other people in similar conditions. The following tables (Table 2 and Table 3) record the answers gathered from the first and second question and their frequencies, ranked from greater to lesser frequency, after summing up second possible choices:

What are the most relevant points that one can learn from these data sets? First, that there are a great variety of coping strategies. We have summarized them into six clusters, as shown in Table 4.

The strongest and most evident cluster is “Relationships”, including family, friends, or other people; they account for 45% of those who answered. Put another way, nearly half of all respondents found their greatest support during this difficult time in other people close to them.

A second cluster could be described as “Interior life”. It gathers first and most importantly explicit prayer (17.5%); and includes “interior life (7.5%); “memories” (6.25%); “belief, confidence” (2.5%); and in the second list “positive thinking” (4.3%), or in a minor measure “acceptance” and “hope”. Summing up, these interior attitudes account for more than one-third of the collected answers.

A third cluster can be designated as “Activity”. It gathers “working” (10%) or “a job” (7.2%) and “sport” (2%) and “volunteering” (2%). These account for about 21% of refugees’ opinions.

The fourth cluster is labelled “Therapy” and gathers those who looked for psychotherapy (7.5%) and those who resorted to medication to overcome their anxiety, trauma, or depression (6.25%). We estimate that almost 14% of interviewees have resorted to such means to cope. If one adds to this cluster those who indicate the need to “control emotions” (18%), then the combined estimate for this cluster amounts to much more, and this cluster could be termed “self-therapy” and is related to the second cluster as well, to “positive thinking”.

The fifth cluster is more scattered and could be referred to as “Positive experiences”, like “volunteers met” (12.5%); experiences of “solidarity” (3.2%); and feelings of “safety” (2.1%); again, about 18% of interviewees.

The sixth cluster can be designated as “Good expectations”. It clearly includes “victory in the war” (8.75%); “good news (2.5%); and “hope” (2%). This comes to a total of about 13% of interviewees.

It is obvious from these data that most people were resorting to more than just one coping strategy, something that reveals that a combination of them is probably the best answer in each case. Another important result is that social contacts play a very important role in the process of coping and resilience, together with an intense interior life. Activity plays an important role, too, in these cases. Coping through professional therapeutic means is of lesser importance, and is necessary in several extreme conditions, or when other coping strategies are not enough to tackle serious psychological problems. Not all the coping strategies listed are self-explanatory; family is self-explanatory, while solidarity is not.

Just one testimony by a young mother and doctor is revealing about coping and feeling supported:

“Children probably help the most. If I did not have children, if I were not supposed to think about them, I might have fallen into a deep depression. Polish people helped a lot when we first arrived in Poland. Someone brought a pot of soup, someone brought a pot of bigos, one guy brought a cheesecake, someone helped financially. And they were all complete strangers. So, that’s how it helped me not to fall into a deep depression”.(43 years old, mother of 2)

### 3.3. Relationship with Host and Volunteers

Two questions were addressed to the interviewees to explore their experience with the people who met and helped them in the process of reception to the host country and the following weeks: “How would you describe the persons you met here?” and “What would you ask to the people that host you?” Our interests were (1) to better understand the complex relationships that can arise between refugees and populations that offer accommodation and help, (2) to better describe dynamics of altruism in these emergency cases, and (3) to know which attitudes and assistance become more useful for people in extreme need.

Regarding the first question, the data clearly describe the same pattern, after building a scale of three levels and codifying the answers into them: “very nice” (66%), “quite nice” (30%), and “less nice” (3%). Many testimonies agree about the goodness and friendly treatment that Ukrainian refugees encountered, especially in Poland, after the hardship of crossing the border with the long waiting times and the harsh conditions during their long travel.

Now, looking to the second interview item, which asked the refugees about what they need or expect from their hosts, Table 5 offers the list of codified answers:

From this list, one can deduce that a quarter of interviewees were already satisfied and did not need anything else from their hosts. This data can be understood as a sign of recognition for all they have already received. Some other demands go in a similar direction: “patience” (9.6%) was intended in several cases as a suggestion and less as a reproach; rather as an acknowledgement of how difficult it could be sometimes to deal with refugees and their demands and needs. “Kindness” and “appreciation” (8.6%) reveal that those immaterial attitudes are needed more than other practical things, among them “more information” appears as most looked-for form of assistance (11.7%). Gratitude emerges several times as the feeling towards hosts and helpers. Those data reveal, in broad terms, that a good relationship and even empathy had grown between refugees and hosts or helpers, an attitude of mutual understanding and affinity. Some testimonies can help to obtain a better sense of the mood reigning among refugees after some time living in their new country:

“People are very open, ready to help. Even if they do not speak Ukrainian or Russian, they are always trying to understand, trying to help. Basically, we were not denied anywhere, and have not faced rudeness anywhere. Everyone was always trying to help. If they can’t help themselves, they refer you to someone who can help you”.(43 years old, mother of 2, refugee in Poland)

“All the people who I met here are my personal heroes. Not only Polish people but all people who I meet here help me. Even with complete strangers who do not know my personal story (I do not like sharing it) I can have close and warm communication. Considering the historical side of Ukrainian-Polish relationship, I can say that what was in the past is left in the past. Indeed, we are brother nations”.(29 years old, mother of 2 children, refugee in Poland)

“We currently have everything we need. Therefore, I would not ask for anything, but only sincerely thank everyone—both our volunteers and the Polish. There are no limits to my gratitude!”.(35 years old, mother of 1 daughter, refugee in Poland)

### 3.4. Feelings and Expectations

Again, two open questions were asked the refugees in our sample: “How do you feel now?” and “Which are your main expectations and hopes?” After coding the many answers and comments to the first question, the results in Table 6 are revealing about such emotional states.

As can be appreciated, over one-third of interviewed refugees still feel bad, other are experiencing mixed feelings, and less than one-fifth feels fine. The distress persists in a consistent proportion of refugees. But for many others, they feel very relieved to be in a secure environment, but nevertheless clearly are missing their homes and the family members left behind, especially husbands.

A similar impression can be perceived from the second question about expectations: 73.4% nourished two related expectations: the end of war and the possibility to return home to meet again their loved ones. Many respondents added that they expected the victory of their country in a war that they felt was unjust. This expectation was even, for many, a hope and a force to cope in their adversity. Only 3% were showing low expectations. This is an important point: the Ukrainian refugees in their great majority do not look for a long-time resettlement in another European country or as an opportunity to find good jobs and to enjoy benefits. They want to return as soon as possible to their own home. Indeed—according to our own data and experience—many of them, after noticing a relative calm environment in the areas where they lived, decided to return by the route they undertook 2 to 3 months earlier. The following selected testimonies speak clearly in that sense:

“I hope that war will end soon, and I will be able to hug my husband”.(29-year-old mother of 2, refugee in Poland)

“I only expect to return home to my husband in peaceful Ukraine as soon as possible”.(32-year-old mother with daughter, refugee in Poland)

“I hope that the war will end, and my family and I will return home to my homeland. And that no one will attack us anymore”.(48-year-old woman, refugee in Poland)

### 3.5. The Role of Religion and Prayer

We gained some insight into the importance of religion and prayer as coping strategies as we reviewed the questions on coping strategies. One can obtain a more complete view when looking at the last item from our questionnaire, in which we asked to what extent prayer was helpful in this difficult time. According to the coded answers, 56.4% prayed more during this time; 16% prayed more or less as they did before the war; and 18% did not pray at all. From these answers, we conclude that prayer has played an important role in coping with these difficult circumstances for more than three-quarters of those surveyed; and more than half recognise that they were praying more. In many cases, praying was the main coping mechanism when all the practical issues could be fixed and when it was expected to have the war end and to return home. Prayer could have a broad reach of intention and could be addressed for additional motives: for winning the war, for the safety of relatives and soldiers, for the security of children, and for overcoming fear and anxiety. One needs to consider that since the majority of refugees are Christian Orthodox and are moving to Catholic countries with different and hard-to-understand languages, these refugees have much less opportunities to attend religious services in their own church and rite, which has clearly limited their religious coping capacities and resources.

Several testimonies can better explain such feelings or views on prayer:

“I pray constantly, now much more, I made sure that prayer helps a lot”.(64 years old, with one daughter and one granddaughter, refugee in Poland)

“Yes, I prayed before and now I pray. They prayed constantly during the bombings. Now that I have free time, I try to dedicate it to God”.(39 years old, mother of 3 children, refugee in Poland)

“Yes. I pray always. Since the beginning of the war, I started to pray more often to keep a spiritual connection with God”.(18 years old woman with 11 years old sister, refugee in Poland)

“I cannot say that I started praying more but I began to feel the need to pray more often. When you read this terrible news about the children, men who are disappearing, my soul hurts and I want to pray”.(34 years old, mother of 2 children, refugee in Poland)

“I have been praying for my entire life. My children have also been praying since childhood. Since the war started, I started praying more. Every time when the sirens are on in my city, I start praying”.(36 years old, mother of 3 children, refugee in Poland)

An interesting feature arising from a correlation test among scalable variables in our coded dataset was that prayer was moderately and positively correlated with the scale on feeling (r = 0.284; *p* ≤ 0.05). It was also correlated with age (r = 0.229; *p* ≤ 0.05). An explanation of the former is that those praying more could feel better or find more consolation and resilience. No other correlations appear as significant in this test.

## 4. Discussion and Conclusions

The described survey was aimed at better understanding coping strategies among refugees of the Ukraine war and how they contribute to achieve resilience. The data here exposed reveal interesting features and invite further reflection.

Even if the sample is limited, it is much greater than other usual qualitative surveys based on semi-structured interviews. This sample is well distributed in four very different locations, with the majority of them collected in Poland, which has a long border with Ukraine and was the main entry point in the European Union for most war refugees. The subjects were chosen at random and found in meeting points for refugees and were a less convenient sample.

One question arising from this research is to what extent it focuses more on coping and how it connects with resilience. The literature on this topic allows for using both terms in continuity: successful coping leads to resilience, and resilience becomes the outcome from good coping strategies. In any case, we can hardly disentangle one from the other.

The present research has applied a qualitative method, which is more fitting for the objectives our team had in mind, but at the same time it developed an analytic approach of text content based on a rather ethnographic hermeneutic approach in its analysis of the collected testimonies. This method allows for a better and in-depth understanding of the questions at stake, but it can suffer from the flexible and subjective criteria of those who undertake such analyses.

The obtained data clearly point to a plurality of coping and resilience strategies in those refugees. We must even speak about different coping styles: for example, those resorting more to prayer and religious means, compared to those who were more focused on professional therapies. In any case, accompaniment and communication appear as the main means of coping in these hard circumstances, much more than agency or psychological treatments. Indeed, if some practical conclusion can be inferred from the present data, it is that promoting resilience needs to resort to enforcing family links and friendship through good means of communication: for example, caring for the adequate functioning of telephones and other platforms that allow refugees to keep in touch with their loved ones. Good relationships with those who help the displaced population mean a lot for those who suffer deprivation from other contacts.

Furthermore, it is important to stress the role that religion and prayer plays in such coping processes, but we are not sure to what extent such data can be translated into practical strategies. It is apparent that religion becomes a sort of “capital”, or even an “affordance”, that can hardly be improvised, but needs training and skill [24]. Several collected testimonies go in that direction. People praying now were praying before the war; perhaps they were praying more after the shelling and alarm sirens started. But they did not learn to pray in the last weeks or months. This was an ability they learned and practiced for a long time prior to the war, and that has now become crucial to achieve a calmer state of mind and to gain resilience and hope. Practice, more than just belief or affiliation, is key to that resilience ability.

It is good to appeal for more professional therapeutic attention for those refugees, as a recently published paper does (Bouchard et al. 2022). However, it is much better to care for the social networks of refugees, and perhaps to offer chaplains and religious services for those who need and demand them. In any case, it is probably true that resilience is the result of a convergence of different factors (Ungar and Theron 2020). This point is confirmed by the testimonies that bear witness to the excellent and even heroic attitudes of those who attended the refugees at their arrival in their destination countries. In other words, resilience works much better when the refugees meet people of great good will, when they can keep their contacts or communication with their loved ones, and when they can cultivate an intense interior life, of which traditional prayer is the best example.

## Figures and Tables

**Table 1 ijerph-19-13094-t001:** Company during exodus.

Condition	Frequency	Percentage
Alone	10	10.6
With one child	15	16
With more children	15	16
With children and more family members	37	39.4
With wife/husband	5	5.3
With friends	5	5.3
No answer	7	7.4

**Table 2 ijerph-19-13094-t002:** Coping strategies.

Coping	Frequency	Percentage
family	27	33.75
prayers	14	17.5
volunteers	10	12.5
friends	9	11.25
working	8	10
victory	7	8.75
interior life	6	7.5
therapy	6	7.5
memories	5	6.25
pills	5	6.25
cannot handle	3	3.75
solidarity	4	3.2
schooling	2	2.5
belief, confidence	2	2.5
news	2	2.5
fury, despair	1	1.25
sport	1	1.25
reading, writing	1	1.25
travel expectations	1	1.25
Total	80	85.1
No answer	14	14.9

**Table 3 ijerph-19-13094-t003:** More coping strategies.

Coping	Frequency	Percentage
control emotions	17	18.1
family attention	15	16.0
praying	10	10.6
a job	7	7.4
social support	6	6.4
positive thinking	4	4.3
integrating	3	3.2
volunteering, helping	2	2.1
acceptance	2	2.1
sports	2	2.1
pills	2	2.1
safety	2	2.1
concern for soldiers	1	1.1
hope better future	1	1.1
listen the youth	1	1.1
psychologist	1	1.1
Total	76	80.9
No answer	18	19.1

**Table 4 ijerph-19-13094-t004:** Main coping strategies ranked by incidence.

Coping Strategy	Incidence (Percentage)
Relationships	45
Interior life	38
Activity	21
Therapy	14
Positive experiences	18
Good expectations	13

**Table 5 ijerph-19-13094-t005:** Demands to host.

Asking	Frequency	Percentage
got all that was needed	12	12.8
more information	11	11.7
nothing	10	10.6
finding a job	9	9.6
patience	9	9.6
better accommodation	6	6.4
prayer	4	4.3
kindness	4	4.3
offer appreciation	4	4.3
psychologist	2	2.1
vital things	2	2.1
medical assistance	1	1.1
emotional support	1	1.1
Total	75	79.8
No answer	19	20.2

**Table 6 ijerph-19-13094-t006:** Feelings.

Feeling	Frequency	Percentage
very bad	35	37.2
more or less well	31	33.0
fine	18	19.1
Total	84	89.4
No answer	10	10.6

## Data Availability

The data collected for this study are available at discretion of the corresponding author.

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
