# Peer review of "Coping and Resilience Strategies among Ukraine War Refugees"

_ijerph, 2022, doi:10.3390/ijerph192013094_

Round 1

Reviewer 1 Report

This article is very topical, pertinent and has enormous potential. However, in my opinion, some changes could be made to strengthen it. I leave my suggestions:

1.Make the introduction more succinct and cohesive. To this end, I suggest that the state of the art/brief conceptual framework be removed from the introduction and constitute an autonomous section. Conversely, it seems to me that it would make more sense the objectives of the study to be included in the introduction. It does not seem very appropriate to refer to these objectives in the methodological section.  

2. The conceptual framework could be a little more developed, especially regarding the health and well-being of refugees and the role of religion in promoting their resilience skills. Moreover, it is crucial to make even more evident the decisive role of the social support provided to the refugees in strengthening their resilience. 

3. I leave some bibliography that may help to strengthen the conceptual framework of the article:

https://journals.sagepub.com/doi/abs/10.1177/0192513X20911068

https://lup.lub.lu.se/search/publication/3008487

https://psycnet.apa.org/record/2020-60959-001

https://journals.sagepub.com/doi/abs/10.1177/1049732316665348

https://journals.sagepub.com/doi/abs/10.1177/0011000020970522

https://go.gale.com/ps/i.do?id=GALE%7CA597092633&sid=googleScholar&v=2.1&it=r&linkaccess=abs&issn=25741306&p=AONE&sw=w&userGroupName=anon%7E96de4384

https://link.springer.com/chapter/10.1007/978-3-030-42303-2_9

https://www.researchgate.net/profile/Pedro-Silva-13/publication/349520642_Quest_for_refuge_Reception_responses_from_the_Global_North/links/61e7faafc5e3103375a6e78a/Quest-for-refuge-Reception-responses-from-the-Global-North.pdf

https://link.springer.com/chapter/10.1007/978-3-030-56650-0_5

https://www.degruyter.com/document/doi/10.1515/spircare-2018-0065/html

4. The testimonies of the refugees must be identified in the body of the text using inverted commas (for example). Moreover, it is important to personalise the testimonies a little more by making reference to the age, city of origin, etc. of the persons who gave the testimonies (like is made - only -  on page 10)

Author Response

We are very grateful for the suggestionns made by this reviewer, especially for the rich list of entries to incorporate in our refrences list.

We have tried to addreess all the queries and to adapt the new text. We address directly the raised issues in the attached file

Reviewer 2 Report

The authors are to be commended for conducting this study for bringing to the social science community the lived experiences of Ukrainians given the current war, forced displacement, and resettlement in the most trying and difficult circumstances since the war and invasion started earlier this year.

As per the manuscript, page 2, in the discussion of the exhaustive bibliographic database, PubMed,  the authors enumerated the number of entries on coping strategies ,etc., were the authors able to identify the number of entries specific to refugees? 

Page 3, line 133, the authors state "several voices have complained..."  Who are these voices- Are there specific researchers who have articulated this? If so, they should be named and cited. 

Materials and Methods: Were the interviewers  provided any training on how to interview? Did they have this type of experience in their background?  The authors mentioned that the interviewers had "enough experience"(page 4, line 193) . Can the authors provide details on what this experience was?

How long on average did the interviews last? Did they range from a certain amount of time to certain amount of time?

Given that the interviews were conducted by Ukranian refugees and with refugees who recently had arrived in Poland, Spain, or Italy, and their experiences were in the very near past, what actions/plans were put in place to address emotions and trauma reactions if they were to exhibited and/or observed in the interviewee?  What if any protocols were in place for the interviewers to receive support if the interviews triggered trauma reactions from them?  If these were part of the ethical protocols established and approved for implementation prior to start of the interviews, they should be described in the discussion of ethical protocols (page 4, starting on line 174). 

Page 4, line 206- give a brief description of the MAXQDA 2022 program-what it is and what its uses are and how it is used here in this study.

The data analysis approach used in this study needs to be clearly described. What analytical technique was used to move from transcripts of the interviews to codes? What theoretical approach guided the analytical technique ( if MAXQDA 2022 figured into this ,the describe ).   

In the scaling of distress, how are "very stressful", "quite stressed", and "low stress" defined?

Was content analysis the analytic approach? Were the results that are  reported  the outcome of content analysis? 

Were there any thematic analysis done on the interviews? 

Is it accurate to state that interviews were read, transcribed, coded, and the presence of each code was counted by reading the transcripts. Then, excerpts (testimonials) from the transcripts were selected that indicated the presence of  of the code to support the section of the code?

In the Table 2, not all the coping strategies listed are self-explanatory, while others are; family is self-explanatory while solidarity is not. Consider providing additional descriptors to provide clearer meaning of the term used. 

This study focused its research questions and data collection more so on coping strategies than resilience strategies. The discussion and conclusion section should be reviewed to make sure that they speak more to the results of the study which were centered around coping strategies, and not make broad associations with resilience, unless the data indicates it. 

The authors should also discuss the strengths and limitations of the methodology and analytic approach.

Overall ,this is important content and findings, therefore, the method used should be clearly described and its strengths and limitations elaborated. 

Author Response

We are very grateful for the suggestions and the questionns raised by the reviewer. The new version tries to address all them and as a result the text is much improved and clearifies aspects which could be less expllicit or somewhat neglected.

You find in the attached file how we addressed directly each query and suggestion, incuding the testimony of the main intervieweer.

Reviewer 3 Report

Line 16 - How many interviews were done? Lines 19, 20 and 231 - Cite the numbers in percentage (it is easier to figure out immediately the importance of the information) Line 36 - Source? Line 137 - The team is the list of the authors of the manuscript? Lines 417 up to 427 - It is necessary to use italics or other resource to highlight that those sentences are from the interviewed people

Author Response

Thank you for appreciating our research. We have try to address your queries

Round 2

Reviewer 2 Report

Page 5, line 207-208

Consider revising to :

All four interviewers were selected after it was assessed that their own experiences and emotions associated with being a war refugee would not compromise their ability to interview other war refugees.    

Author Response

We propose this new redaction: 

The interviewers were selected among Ukraine war refugees, who because of their ethnic affinity with the interviewees and sharing similar experiences, could have an advantage to empathise with them, and thus to be in a better position to address the stress and emotional trauma that could emerge during the interviews.